# Branch error reduction criterion-based signal recursive decomposition and its application to wind power generation forecasting

Fen Xiao[1], Siyu Yang[1], Xiao Li[2]*, Junhong Ni[2]

1 State Grid Fujian Electric Power Co., LTD., Fuzhou, China, 2 North China Electric Power University, Baoding, China

* lixiao_ncepu@163.com

## Abstract

Due to the ability of sidestepping mode aliasing and endpoint effects, variational mode decomposition (VMD) is usually used as the forecasting module of a hybrid model in time-series forecasting. However, the forecast accuracy of the hybrid model is sensitive to the manually set mode number of VMD; neither underdecomposition (the mode number is too small) nor over-decomposition (the mode number is too large) improves forecasting accuracy. To address this issue, a branch error reduction (BER) criterion is proposed in this study that is based on which a mode number adaptive VMD-based recursive decomposition method is used. This decomposition method is combined with commonly used single forecasting models and applied to the wind power generation forecasting task. Experimental results validate the effectiveness of the proposed combination.

## Introduction

The increasing frequency of human activities and rapid development of the social economy increase electricity demand, which drives the growth of the global power generation industry. In order to meet future energy and power demand, adapt to changes in energy supply and demand and environmental situation, and realize long-term sustainable development, it is urgent to strengthen the development and utilization of clean and renewable energy [1]. The utilization and exploration of renewable energy power generation will be one of the important issues in the power industry in the future [2]. Because new energy generation is strongly affected by environmental factors, the time series of power generation is intermittent and volatile, which is not conducive to the stable operation and rational planning of the power system [3]. Therefore, it is critical to develop an effective time series model for power generation forecasting.

Currently, time series models can be roughly divided into three categories: classical statistical models, machine learning models, and hybrid models. The ARIMA model is one of the typical representatives of classical statistical models and has been widely used in load forecasting [4]. Commonly used machine learning models include the back propagation (BP) neural network [5], long short-term memory (LSTM) [6], and support vector network (SVM) [7].

Compared to single models, the hybrid model has better performance when solving complex time series forecasting issues. Hybrid models can be further classified into two subtypes.

**Funding:** Funding information: State Grid Fujian Electric Power Co. Ltd.: SGTYHT/20-JS-223 (SGFJJY00GHJS2200054) The funders had no role in study design, data collection and analysis, decision to publish, or preparation of the manuscript.

**Competing interests:** The authors have declared that no competing interests exist.

The first subtype integrates the forecasting results of several single models. Zhang et al. [8] used a genetic algorithm (GA) to optimize the parameters of support vector regression (SVR) in the time series forecasting task. Choi et al. [9] combined CNN and BiLSTM together to handle the strong long memory serial dependence feature of the dataset. The second subtype decomposes the time series into subsignals and sums the forecasting results of the subsignals. Bai et al. [10] decomposed the time series using the wavelet transform (WT) [11] and forecasted future air pollutant concentration measurements with a BP neural network. Zheng et al. [12] and Chen et al. [13] used empirical mode decomposition (EMD) [14] to decompose the electric load and combined LSTM and extreme learning machine (ELM). Qin et al. [15] combined ensemble EMD (EEMD) [16] with local polynomial prediction (LPP) as the final model for the forecasting task. Lv et al. [17] decomposed time series using VMD and used LSTM to forecast power load. Cai et al. [18] proposed a combination of VMD, gated recurrent unit (GRU) and time convolutional network (TCN) to achieve a satisfactory power load forecasting result.

The time series of electricity generation is generally a broadband signal, and its future trend is not stable. Therefore, it is difficult to approximate the relationship between historical measurements and its future changes. The future trend of a narrowband signal is normally considered to be more stable. Therefore, the second type of hybrid model is used to decompose the time series of power generation into narrowband modes, and the final forecasted results are obtained by summarizing the forecasted results of each mode. Among the decomposition methods, WT is nonadaptive, and the selection of the optimal wavelet basis strongly affects the decomposition results. EMD suffers from mode aliasing and endpoint effects. EEMD can overcome the shortcomings of EMD to a certain extent but still requires complex calculations and incompletely neutralizes white noise. VMD [19] is a nonrecursive and robust signal decomposition method and has a solid theoretical basis and circumvents the disadvantages of similar methods.

Therefore, VMD is the best choice for the signal decomposition module in hybrid models. However, VMD is sensitive to the mode number, which must be set manually. A mode number that is too small leads to underdecomposition of the signal, while a mode number that is too large results in overdecomposition. Both underdecomposition and overdecomposition decrease the forecast accuracy of the hybrid model. Therefore, it is important to adaptively determine the optimal mode number for VMD. In many studies, the mode number was adaptively aligned with the number determined by EMD [20–22], but this method cannot mitigate the negative impact of modal aliasing on forecast accuracy. Huang et al. [23] used a genetic algorithm to optimize VMD parameters to reduce the decomposition loss, but the addition of a new algorithm makes the problem complex and inefficient.

To address this issue, we design a branch error reduction criterion, upon which a VMD-based recursive decomposition method with adaptive mode number is proposed.

The primary contributions of this study are as follows:

- The BER criterion is designed, and we show that a subsignal that is further decomposed leads to better forecasting accuracy if this subsignal satisfies the BER criterion.

- A mode number adaptive VMD-based recursive decomposition method is proposed. The hybrid model that combines the proposed decomposition method and commonly used forecasting single model is used to fulfill the wind power generation forecasting task.

- Experimental results validate that the proposed VMD-based recursive decomposition method can effectively extract the fluctuation patterns of wind power and improve the forecast accuracy.

The remainder of this paper is organized as follows: Principles of VMD and permutation entropy; Derivation of the branch error reduction criterion and the VMD-based recursive decomposition method; Experiments that verified the effectiveness of the proposed method; Summary of the paper.

## VMD and permutation entropy

### VMD

VMD assumes that all components are narrowband signals concentrated around their respective center frequencies; thus, VMD constructs constrained optimization problems based on narrowband conditions of components to estimate the center frequency of subsignals and reconstruct corresponding components [12].

Under the assumption that the original signal $f(t)$ is decomposed into $K$ subsignals and the decomposition sequence is guaranteed to be a mode component with a finite bandwidth around a central frequency, the sum of the estimated bandwidth of each mode is minimized, and the constraint is that the sum of all modes is equal to the original signal. Then, the variational model can be formulated as:

$$\begin{cases} \min\limits_{\{u_k\},\{\omega_k\}} \left\{ \sum\limits_k ||\partial_t \left[ \left(\delta(t) + \frac{j}{\pi t}\right) * u_k(t) \right] e^{-j\omega_k t} ||_2^2 \right\} \\ s.t. \sum\limits_k u_k(t) = f(t) \end{cases} \tag{1}$$

where $u_k(t)$ is the mode function; $\omega_k$ is the mode center frequency; $K$ is the number of modes; $\delta$ is the Dirac function; $*$ is the convolution calculator; and $f(t)$ is the input signal.

The Lagrange multiplier $\lambda(t)$ and the quadratic penalty factor $\alpha$ are introduced to transform the constrained algorithm into an unconstrained variational problem:

$$L(\{u_k(t)\}, \{\omega_k\}, \lambda(t)) = \alpha \sum\limits_k ||\partial_t \left[ \left(\delta(t) + \frac{j}{\pi t}\right) * u_k(t) \right] e^{-j\omega_k t} ||_2^2$$

$$+ ||g(t) - \sum\limits_k u_k(t)||_2^2 + \left\langle \lambda(t), g(t) - \sum\limits_k u_k(t) \right\rangle \tag{2}$$

The optimal solution of the variational problem is obtained by iteratively updating $u_k^{n+1}(t)$, $\omega_k^{n+1}(t)$ and $\lambda_k^{n+1}(t)$ using the alternating direction method of the multipliers. In this study, the iterative process of the Fourier transform of $u_k(t)$, $\omega_k$ and $\lambda(t)$ can be expressed as:

$$\hat{u}_k^{n+1}(\omega) = \frac{\hat{f}(\omega) - \sum\limits_{i \neq k} \hat{u}_i(\omega) + \frac{\hat{\lambda}(\omega)}{2}}{1 + 2\alpha(\omega - \omega_k)^2} \tag{3}$$

$$\omega_k^{n+1}(\omega) = \frac{\int_0^\infty \omega |\hat{u}_k(\omega)|^2 d\omega}{\int_0^\infty |\hat{u}_k(\omega)|^2 d\omega} \tag{4}$$

$$\hat{\lambda}^{n+1}(\omega) \leftarrow \hat{\lambda}^n(\omega) + \eta[\hat{f}(\omega) - \sum\limits_k \hat{u}_k^{n+1}(\omega)] \tag{5}$$

where $\eta$ is the noise tolerance of the signal.

**Table 1. Decomposition results of daily power generation data in 2020 under different mode numbers.**

| Model | 6 | 7 | 8 | 9 | 10 | 11 | 12 | 13 |
|-------|---|---|---|---|----|----|----|----|
| ELM | 26906.86 | 17269.87 | 9553.03 | 12591.53 | 37514.39 | 77360.43 | - | - |
| LSTM | - | - | 23629.83 | 1300.68 | 809.20 | 490.02 | 1439.95 | 1803.80 |
| LSSVM | - | 370.82 | 237.62 | 208.83 | 195.28 | 176.50 | 153.20 | 159.09 |

As a decomposition algorithm widely used in signal processing, VMD effectively overcomes the problems of mode aliasing and endpoint effects; thus, it is often combined with forecasting models to form hybrid models for time series forecasting. The number of decomposition modes is a key parameter that must be set manually when using VMD, which is critical to the forecasting results. For example, daily power generation is the result of multiple factors, and its time series is a combination of several modes with different vibration frequencies. Underdecomposition of daily power generation series cannot accurately separate each mode, resulting in the overlap of modes with different fluctuation patterns, which affects the accuracy of the final results, while in the case of overdecomposition, the increase in the mode number corresponds to the growth of computation and training time, which damages or even cancels the advantage of the stable future trend of some subsignals. Therefore, the effective determination of the optimal mode number of VMD becomes a critical problem to solve. Table 1 shows the final forecasting errors obtained by decomposing the daily power generation data in 2020 according to different mode numbers under the three single forecasting models.

Table 1 shows that the forecasting error varies strongly with an increase in the number of modes, which highlights the importance of the choice of the mode number for accurate forecasting. The trend of the variation in the normalization error corresponding to the three hybrid models is shown more intuitively in Fig 1, which means that the optimal mode number differs for different forecasting models, and this parameter cannot be derived empirically or by simple data processing. Therefore, a method that can effectively determine the optimal mode number corresponding to different forecasting models is important to develop.

## Permutation entropy

During the process of VMD signal decomposition, a residual fraction (RF) will be generated that contains more random noise but may also have part of the information of the original

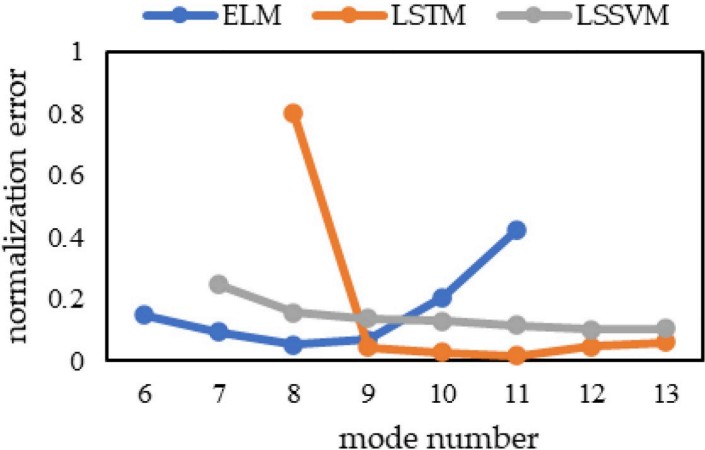

**Fig 1. Normalization error trends at different mode numbers.**

series; thus, the permutation entropy (PE) criterion is considered as the basis for filtering the residual component in this paper. PE was proposed by Bandt et al. [24] in 2002 to detect the randomness of time series and is suitable to analyze nonstationary signals with good robustness. The entropy of a signal determines its random degree: the larger the entropy, the more random the signal is. Therefore, RF can be detected by PE. The calculation steps are as follows.

We denote a time series as $\mathbf{S} = \{s(1), s(2), \cdots, s(n)\}$ and obtain the matrix after space reconstruction:

$$\mathbf{S}' = \begin{bmatrix} s(1) & s(1+\tau) & \cdots & s(1+(m-1)\tau) \\ \vdots & \vdots & \ddots & \vdots \\ s(i) & s(i+\tau) & \cdots & s(i+(m-1)\tau) \\ \vdots & \vdots & \ddots & \vdots \\ s(n-(m-1)\tau) & s(n-(m-1)\tau) & \cdots & s(n) \end{bmatrix} \tag{6}$$

where $\tau$ is the delay time and $m$ is the embedding dimension. Rearranging the reconstructed matrix in ascending order yields:

$$s(i+(j_1-1)\tau) \leq s(i+(j_2-1)\tau) \leq \cdots \leq s(i+(j_m-1)\tau) \tag{7}$$

where $j_1, j_2, \cdots, j_m$ are the index values of elements in the reconstructed component.

For any segment $s_i$, a set of symbol sequences $\{j_1, j_2, \cdots, j_m\}$ can be obtained; thus, there are different symbol sequences mapped from dimensional space. Calculating the probability of each symbol sequence, PE can be defined as:

$$H_P(m) = -\sum_{j=1}^{N-(m-1)\tau} P_j \ln P_j \tag{8}$$

where PE reaches its maximum $\ln(m!)$ when $P_j = 1/m!$. In the real process, normalization is usually performed:

$$0 \leq H_P = H_P/\ln(m!) \leq 1 \tag{9}$$

PE thus describes the randomness of the time series. By calculating the entropy of RF, the components with a larger proportion of noise are eliminated to reduce random noise.

## VMD-based signal recursive decomposition

Based on the shortcomings of VMD, we propose a mode number adaptive VMD-based recursive decomposition method, which is expected to automatically calculate the corresponding optimal mode number when combining different forecasting models.

### Branch error reduction criterion

BER uses the mean absolute error (MAE) as an expression of test error and determines whether further decomposition is required. If the sum of the subsignals' MAE is less than the MAE before decomposition, the decomposition is meaningful.

The criterion is based on the following theorem:

Theorem 1. The total testing error decreases when the sum of the errors of the child branches is smaller than the error of the parent branch.

Proof:

We assume that $\mathbf{V} = [V_1, V_2, \cdots, V_N]$ is testing data, where $V^{(k)}$ is the $k$th subsignal of $\mathbf{V}$ and its MAE is $e^{(k)}$; $V^{(k,q)}$ is the $q$th subsignal of $V^{(k)}$ and its MAE is $e^{(k,\,q)}$. If the testing error before and after decomposition satisfies:

$$\sum_q e^{(k_0,q)} < e^{(k_0)} \tag{10}$$

then it can also be expressed as:

$$\frac{1}{N} \sum_{q,t} \left| \hat{V}_t^{(k_0,q)} - V_t^{(k_0,q)} \right| < \frac{1}{N} \sum_t \left| \hat{V}_t^{(k_0)} - V_t^{(k_0)} \right| \tag{11}$$

We thus extend Eq (11) as:

$$\frac{1}{N} \sum_{q,t} \left| \hat{x}_t^{(k_0,q)} - x_t^{(k_0,q)} \right| + \frac{1}{N} \sum_{k \neq k_0,t} \left| \hat{V}_t^{(k)} - V_t^{(k)} \right|$$
$$< \frac{1}{N} \sum_t \left| \hat{V}_t^{(k_0)} - V_t^{(k_0)} \right| + \frac{1}{N} \sum_{k \neq k_0,t} \left| \hat{V}_t^{(k)} - V_t^{(k)} \right| \tag{12}$$

Expressing Eq (12) in another form:

$$e = \sum_q e^{(k_0,q)} + \sum_{k \neq k_0} e^{(k)} < e^{(k_0)} + \sum_{k \neq k_0} e^{(k)} = \sum_k e^{(k)} \tag{13}$$

where $\hat{V}_t^{(k)}$ and $\hat{V}_t^{(k_0,q)}$ are the forecasting values of and, respectively.

This derivation shows that if Theorem 1 is satisfied, the total testing error after decomposition is reduced, which proves that the decomposition is meaningful.

## BER-based decomposition

Based on Theorem 1, we propose a recursive signal decomposition method based on BER. The specific decomposition process of this method is as follows.

Step 1: The original time series is decomposed with VMD at the first level, the number of decompositions is set to, and the subsignals are input into each forecasting model to obtain the corresponding testing error.

Step 2: Decompose and forecast the subsignals in the next level and set the number of decompositions to.

Step 3: Check if the error before and after decomposition satisfies the BER criterion. If the error decreases, the decomposition is retained, and the operation is repeated in Step 2; otherwise, the decomposition is invalid and terminates.

A flow diagram of the decomposition method is shown in Fig 2.

Wind power generation data are decomposed for the first time by VMD, and the mode number $K_0$ is obtained via a simple calculation and is a small value in the range of the optimal mode number of different forecasting models. $K'$ is selected to ensure the initial decomposition of the data without overdecomposition. In this study, is set to 2. In addition, the subsignals of the final output are through different levels of decomposition in most cases.

The hybrid model used in this study is shown in Fig 3. In this model, the BER-based decomposition method is used as the signal decomposition module to first decompose the historical wind power generation data into subsignals adaptively. Then, these subsignals are fed into the

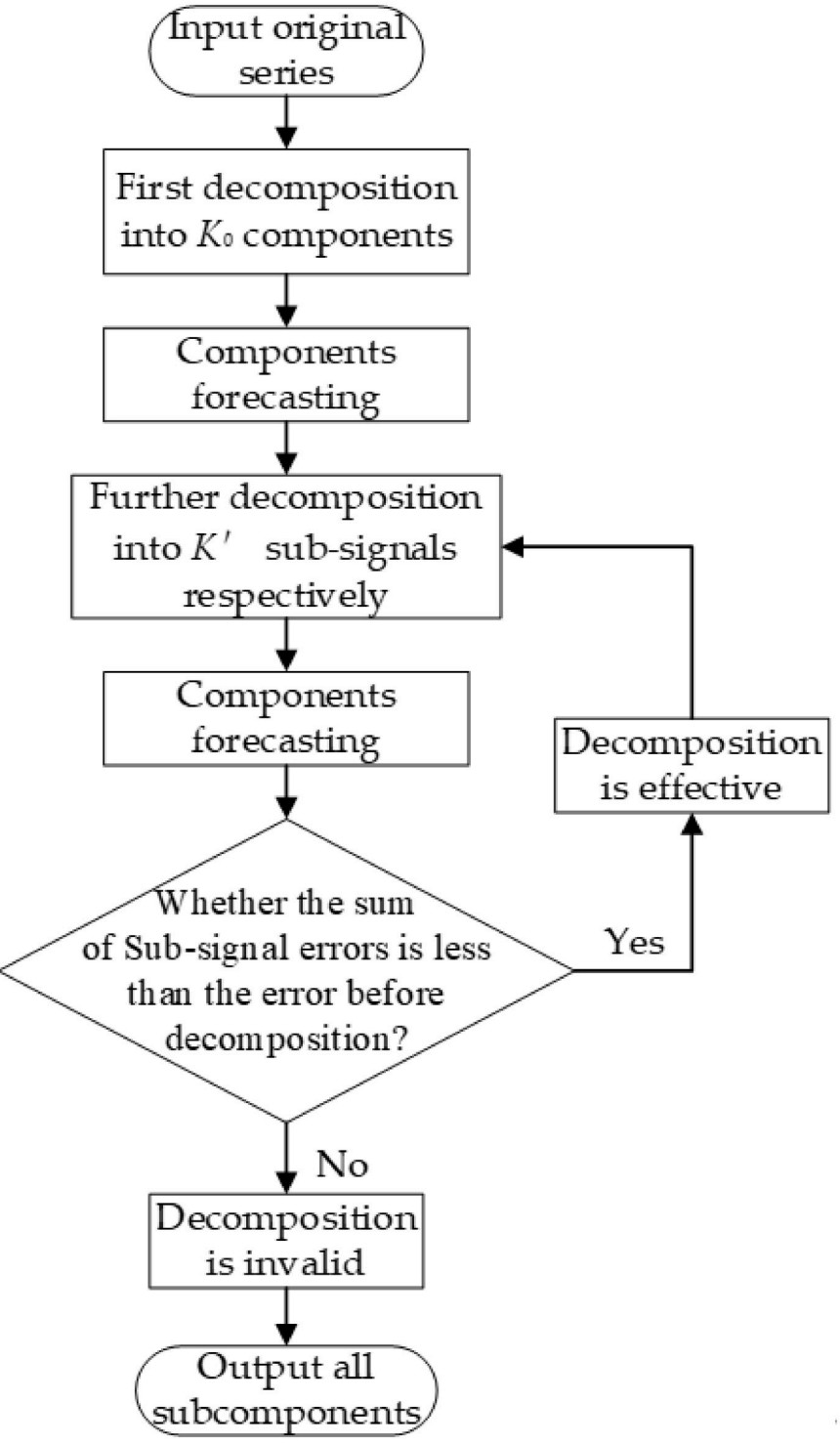

**Fig 2. Flow chart of BER to determine the mode number.**

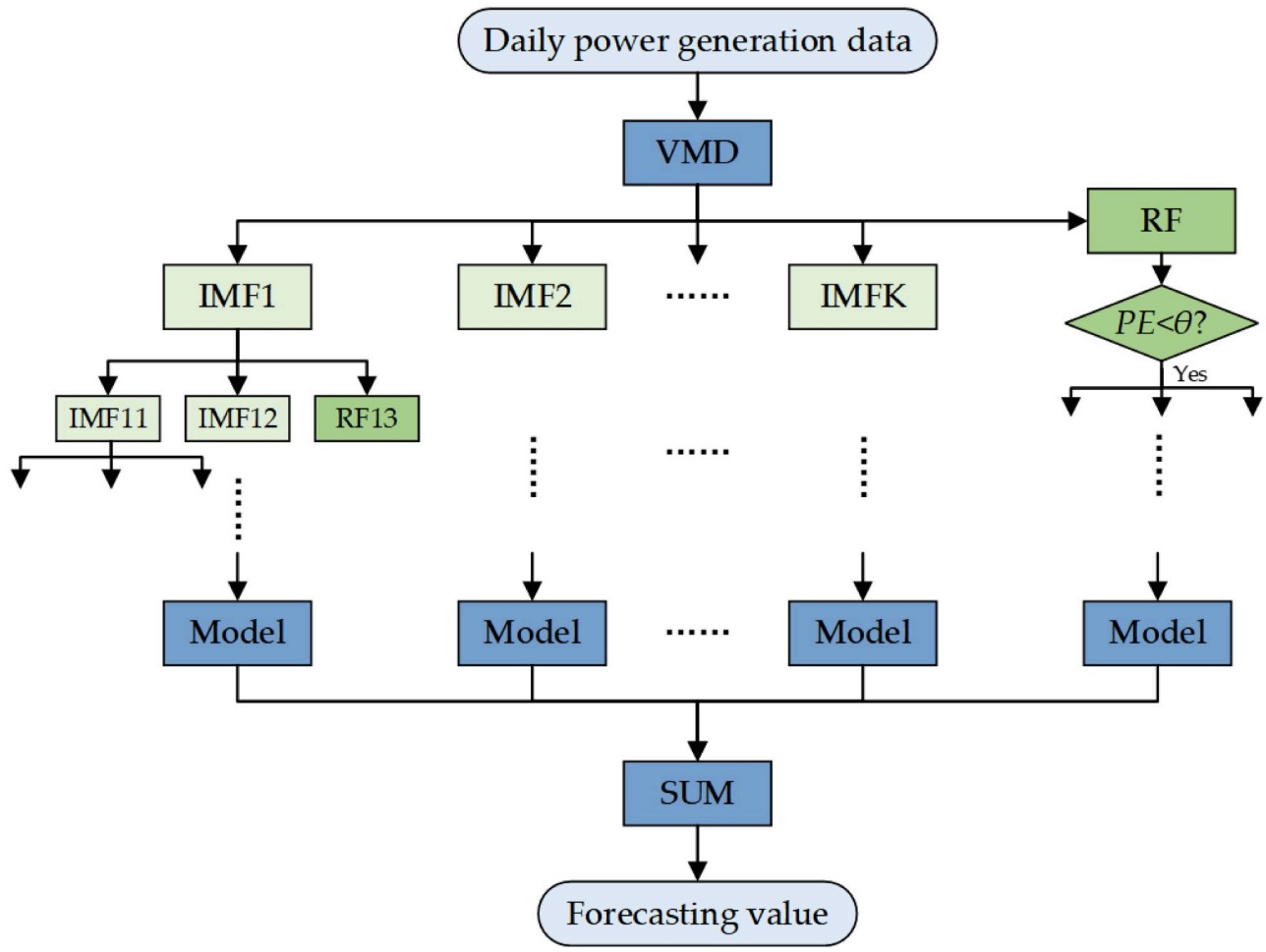

**Fig 3. Flow chart of the proposed method.**

different models to realize the forecasting task, and finally, the forecast results of the subsignals are superimposed to obtain the predicted power generation. In the hybrid models used in this study, the forecasting models can use statistical, machine learning, or deep learning models such as LSSVM, ELM, DBN, LSTM, etc. In addition, VMD decomposes the signal into multiple narrowband components and a residual component, which may contain the high-frequency component of the original signal; thus, directly discarding the residual component may cause the loss of the high-frequency component and affect the forecast accuracy. Thus, we choose PE as a measure of signal randomness and set a threshold $\theta = 1$ to filter the residual components of each decomposition.

## Data experiments

### Power generation datasets

In this example, the historical data of wind power generation in Fujian Province from 2020 to 2021 are selected for the experiment. The sampling interval of this dataset is 1 h, with a total of 731 wind power generation datasets, as shown in Fig 4. The first ten months are selected as the training set, and the last two months are selected as the testing set.

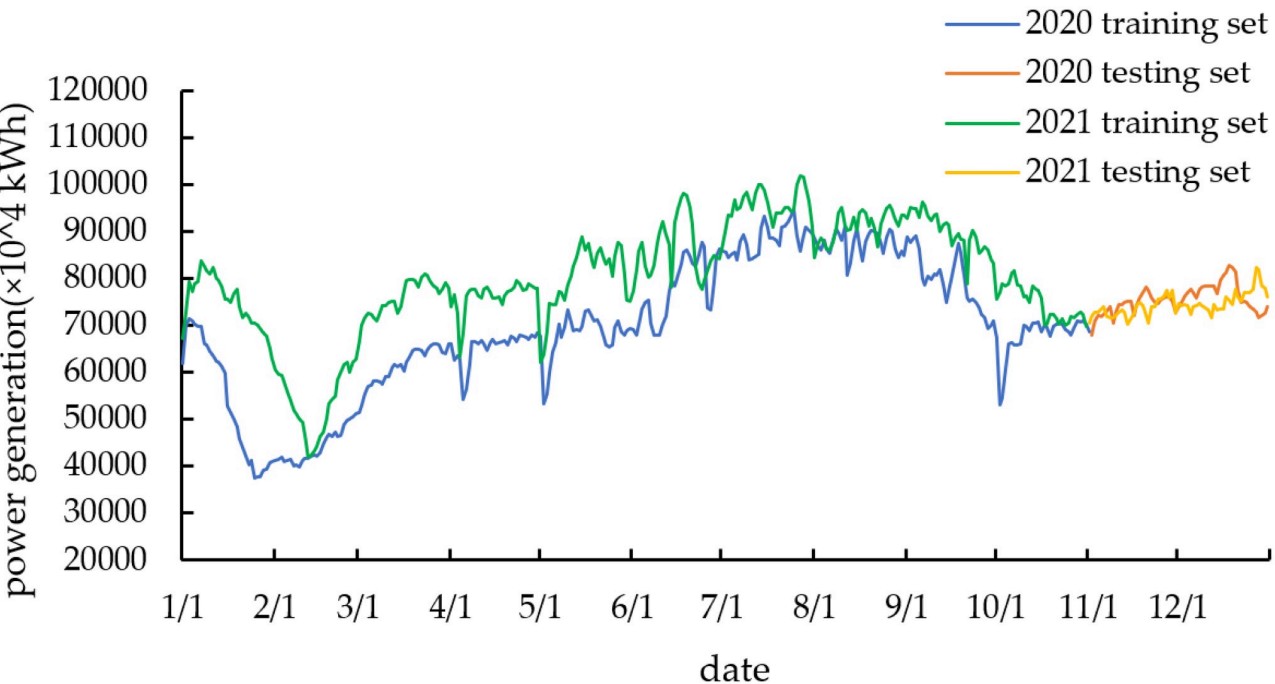

**Fig 4. Trend of wind power generation in 2020 and 2021.**

As shown in Fig 4, the overall wind power generation in Fujian Province increases from 2020 to 2021 and shows a marked seasonal fluctuation. Correspondingly, the stationarity of the series is poor, which can also be seen from the Augmented Dickey-Fuller (ADF) Test. The ADF test results are all greater than critical values, which proves that the series is non-stationary and must be further processed. The statistics of the wind power generation by season are shown in Table 2 and Fig 5, where January-March for spring, April-June for summer, July September for autumn, and October-December for winter.

The wind power generation datasets in 2021 show an overall increase compared to 2020 and are primarily reflected in the three seasons of spring, summer, and autumn; the standard deviation in 2021 is relatively small and stable overall. In addition, the seasonal differences within a year are strong, showing high power generation in summer and autumn, and low

**Table 2. Analysis of the daily power generation datasets.**

| Year | Season | Maximum | Minimum | Average | Standard deviation | Kurtosis | Skewness | ADF |
|------|--------|---------|---------|---------|--------------------|----------|----------|-----|
| 2020 | *Ayear* | 94722.37 | 37482.15 | 70535.15 | 13445.52 | -0.60017 | -0.07689 | -1.53987 |
| | *Spring* | 71441.16 | 37482.15 | 53559.38 | 10556.19 | 0.016509 | -1.50993 | -2.61722 |
| | *Summer* | 87595.2 | 53401.02 | 70531.12 | 7395.71 | 0.582071 | 0.463894 | -1.84112 |
| | *Autumn* | 94722.37 | 52945.79 | 80081.74 | 9622.918 | -0.52217 | -0.7652 | 2.74015 |
| | *Winter* | 82782.82 | 70575.13 | 75600.89 | 2788.615 | 0.576969 | 0.12333 | -1.64434 |
| 2021 | *Ayear* | 101772.5 | 41920.3 | 79122.76 | 11324.93 | -0.56403 | 0.816906 | -2.09905 |
| | *Spring* | 83659.18 | 41920.3 | 68567.24 | 11489.91 | -0.81146 | -0.51751 | -2.49696 |
| | *Summer* | 98046.13 | 62004.29 | 80890.61 | 6974.329 | 0.09976 | 0.783972 | -3.22889 |
| | *Autumn* | 101772.5 | 69698.15 | 87854.12 | 8768.691 | -0.72712 | -0.66269 | -1.82974 |
| | *Winter* | 82359.6 | 70116.29 | 74472.17 | 2405.469 | 1.02896 | 1.781289 | -2.29268 |

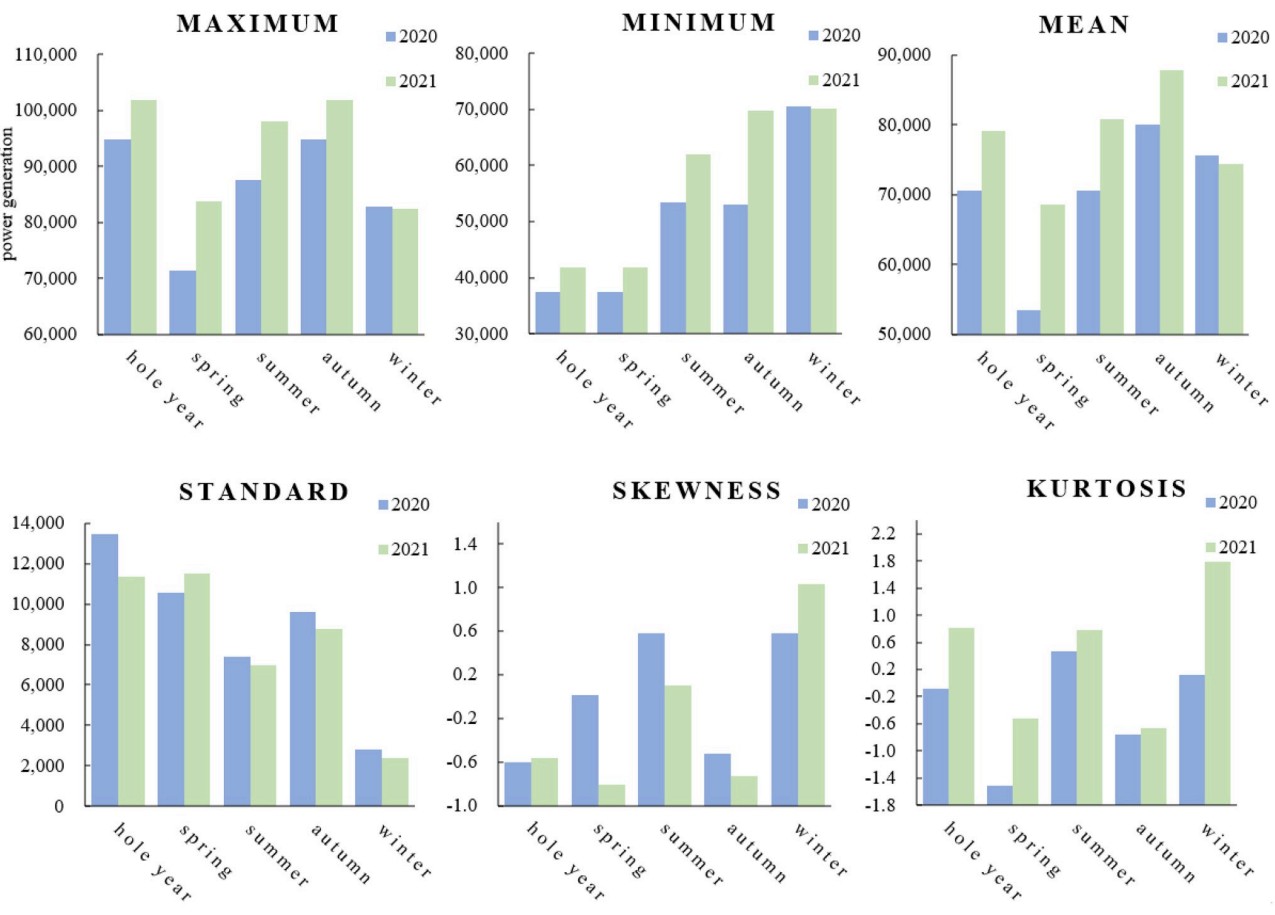

**Fig 5. Statistics of wind power generation datasets.**

power generation in spring and autumn; the standard deviation in winter is maintained at a low level, while the variation in spring is more intense and random.

## Evaluation indicators

**Error indicators.** In this study, the mean absolute error (MAE), root mean square error (RMSE), and mean absolute percentage error (MAPE) are selected as evaluation indicators:

$$MAE = \frac{1}{N}\sum_{t=1}^{N}\left| V_t^a - V_t^f \right| \tag{14}$$

$$RMSE = \sqrt{\frac{1}{N}\sum_{t=1}^{N}\left( V_t^a - V_t^f \right)^2} \tag{15}$$

$$MAPE = \frac{1}{N}\sum_{t=1}^{N}\left| \frac{V_t^a - V_t^f}{V_t^f} \right| \times 100\% \tag{16}$$

where $N$ is the length of the series, $V_t^a$ is the real series, and $V_t^f$ is the forecasting series.

**Improvement indicators.** To evaluate the improvement of the proposed decomposition method compared with other methods, the following formula is used as improvement indicators based on the above error indicators:

$$P = \frac{e_1 - e_0}{e_1} \times 100\% \qquad (17)$$

where $P$ is the percentage of error reduction, $e_1$ and $e_0$ represent the error of the proposed method and comparative experiment, respectively.

## Comparative experiments

**Decomposition according to the center frequency.** There is still a lack of general guidelines for the selection of the mode number [25]. Among the traditional methods of determining the mode number, the more intuitive and simple method is to observe whether the center frequency is aliased [26]. The size of $K$ is increased from $K = 2$ to observe the distribution of the center frequency. The center frequencies for different values of K are shown in Table 3.

Table 3 shows that when the mode number is above 5, the center frequency of the last mode component always remains relatively stable. If $K$ is continuously increased, the more layers of decompostion there are, the smaller the interval of the center frequencies of each component will be, and the more likely it is to generate additional noise components because the center frequency of the last layer remains unchanged. Thus, the optimal value of the mode number for the first decomposition is 5. Fig 6 shows the decomposed signal curves of the daily power generation data in 2020 and 2021, where IMF1-IMF5 are the narrowband components and RF is the residual component. From top to bottom, the curve vibration frequency becomes increasingly intense and irregular.

Similarly, a second decomposition is performed for all components according to the central frequencies to obtain the central frequencies at $K = 2, 3, 4$. The central frequencies obtained by performing this operation for the power generation datasets in 2020 and 2021 are shown in Table 4. The values of $K$ for the second decomposition are 3, 3, 3, 3, 1, and 3 in 2020, and 3, 2, 3, 2, 1, and 3 in 2021. The central frequencies of the fifth component are already aliased at $K = 2$; thus, the second decomposition is not performed.

**Decomposition according to BER criterion.** In a deeper decomposition of the original wind power generation series based on BER, the final decomposition results differ markedly from the decomposition according to the central frequency, and different forecasting models correspond to different optimal decompositions. Taking LSSVM as the forecasting model as an example, the decomposition process of the power generation series in 2020 is shown in Fig 7.

With LSSVM as the forecasting model, four components can be decomposed for the second time, and eleven subsignals can be decomposed for the third time. According to the same decomposition process, the optimal decomposition is performed under the forecasting models

**Table 3. Central frequency of the first decomposition.**

| K | Center Frequency | | | | | | |
|---|---|---|---|---|---|---|---|
| 3 | 0.000022 | 0.0135 | 0.128 | - | - | - | - |
| 4 | 0.000019 | 0.0112 | 0.0429 | 0.1352 | - | - | - |
| 5 | 0.000018 | 0.0112 | 0.0429 | 0.1351 | 0.4217 | - | - |
| 6 | 0.000018 | 0.0111 | 0.0417 | 0.1224 | 0.1828 | 0.4236 | - |
| 7 | 0.000018 | 0.0108 | 0.0374 | 0.0784 | 0.1403 | 0.2747 | 0.427 |

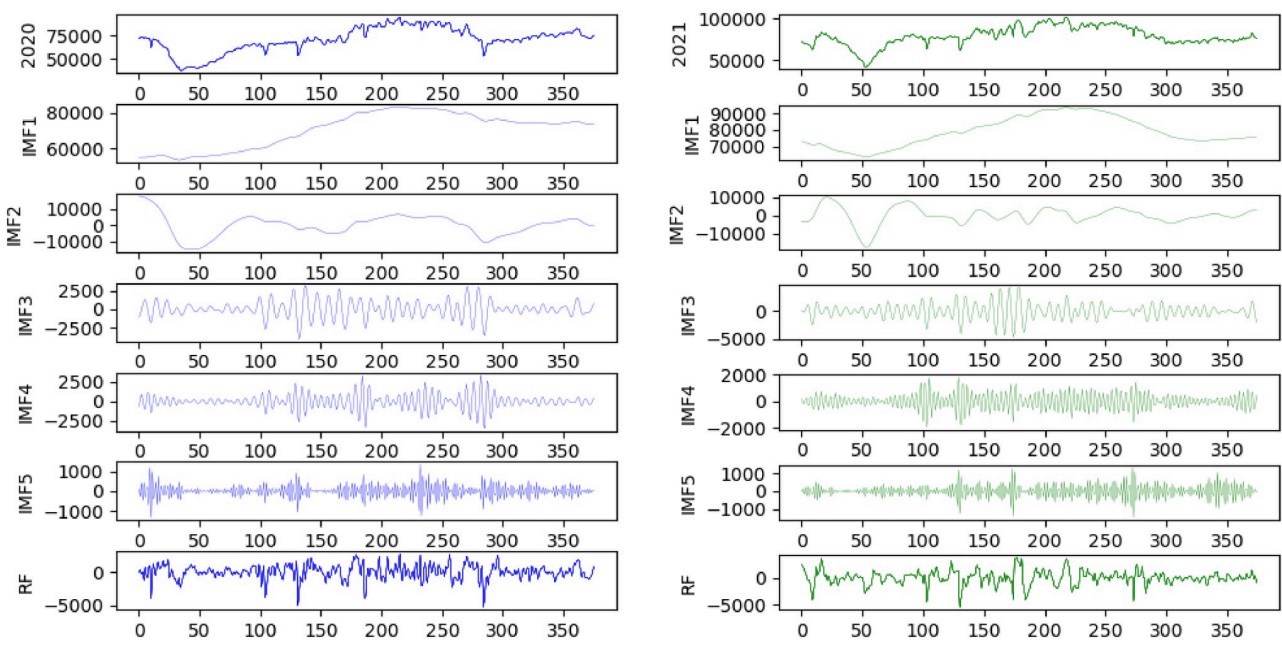

**Fig 6. Normalization error trends at different mode numbers.**

of ELM, LSTM, and DBN, and the percentage decrease of the error sum corresponding to the latter two decompositions is shown in Table 5. With the gradual depth of decomposition, the error sum decreases, satisfying the BER criterion.

From this data, RF is shown to produce the largest prediction variations and testing errors. RF contains more noise but also preserves some information about the original series; thus, we measure whether the RF and the subsignals obtained from the second decomposition of RF should be retained by PE. Table 6 shows the PE of the above components and compares it with the entropy of the five narrowband components acquired from the first decomposition, which is referred to as avg. IMF.

Table 6 shows that although most of the values of several residual components are greater than the avg. IMF, they do not exceed the threshold; thus, no component must be removed. Also, after the second decomposition of the residual components, the PE of the first two

**Table 4. Central frequencies of the second decomposition in 2020 and 2021.**

| Year | Subsignal | K = 2 | K = 3 | K = 4 |
|------|-----------|-------|-------|-------|
| 2020 | 1 | 1.82E-05; 0.0319 | 3.02E-06; 0.0025; 0.0338 | 2.79E-06; 0.0307; 0.0489; 0.0024 |
| | 2 | 0.0092; 0.0301 | 0.0081; 0.0120; 0.0323 | 0.0105; 0.0295; 0.0452; 0.0066 |
| | 3 | 0.1019; 0.1213 | 0.0751; 0.1055; 0.1239 | 0.0750; 0.1051; 0.1214; 0.1394 |
| | 4 | 0.1551; 0.1823 | 0.1517; 0.1753; 0.1944 | 0.1512; 0.1738; 0.1983; 0.1854 |
| | 5 | 0.4105; 0.4298 | 0.3945; 0.4308; 0.4180 | - |
| 2021 | 1 | 9.40E-07; 0.0030 | 8.67E-07; 0.0404; 0.0030 | 5.77E-07; 0.0028; 0.0076; 0.0411 |
| | 2 | 0.014; 0.034 | 0.010; 0.037; 0.018 | - |
| | 3 | 0.114; 0.140 | 0.113; 0.138; 0.161 | 0.095; 0.139; 0.165; 0.119 |
| | 4 | 0.258; 0.287 | 0.251; 0.270; 0.289 | 0.230; 0.271; 0.255; 0.289 |
| | 5 | 0.432; 0.409 | - | - |

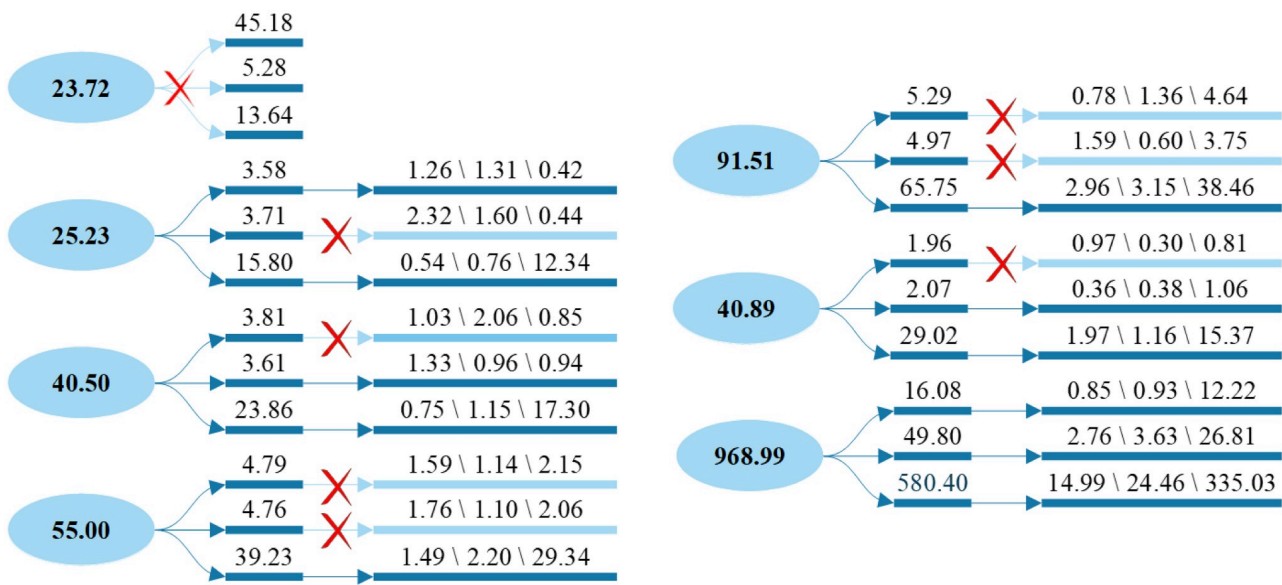

**Fig 7. LSSVM decomposition process.**

secondary components of the decomposition is significantly reduced, tending to the range of the narrowband components, while the PE of the secondary residual components has increased to a certain extent, indicating that the noise components are more obviously separated after the second decomposition. In the experiments, the secondary residual component with the largest value of PE is removed as an independent comparison experiment.

Thirteen comparison experiments of different decomposition patterns are conducted for each forecasting model separately, all of which divided the original series into five narrowband components and one residual component at the first level of decomposition. The experiments are divided into five groups, and the variables between groups are the number of decomposition layers and basis; the variables within groups are the treatment methods of the residual components. In the last group of experiments, a comparison experiment of removing secondary residual components is conducted. Therefore, when comparisons are made between groups, the best-performing type of experimental data within the group is selected in all cases:

**Table 5. Percentage decrease in the error sum.**

| year | decomposition level | LSTM | ELM | LSSVM | DBN |
|---|---|---|---|---|---|
| 2020 | 2 | -5.37% | -31.90% | -29.22% | -33.62% |
| | 3 | -7.57% | -47.41% | -54.91% | -44.49% |
| 2021 | 2 | -28.61% | -35.06% | -32.29% | -11.71% |
| | 3 | -41.67% | -41.55% | -43.87% | -17.93% |

**Table 6. PE of each component.**

| Year | avg. IMF | first decomposition | second decomposition | | |
|---|---|---|---|---|---|
| | | RF | RF1 | RF2 | RF3 |
| 2020 | 0.4479 | 0.6085 | 0.4235 | 0.5337 | 0.6179 |
| 2021 | 0.4451 | 0.6149 | 0.4478 | 0.5421 | 0.6206 |

A. Direct secondary decomposition

- Decomposing the six components according to $K = 2$.

- Decomposing the six components according to $K = 3$.

B. Secondary decomposition according to the central frequency

- Decomposing five narrowband components according to the center frequency, the residual components are not subjected to the second level of decomposition.

- Decomposing five narrowband components according to the center frequency, the residual components are decomposed according to $K = 2$.

- Decomposing five narrowband components according to the center frequency, the residual components are decomposed according to $K = 3$.

C. Secondary decomposition according to the BER criterion

- Five narrowband components are decomposed for a second time according to the BER criterion, and the residual components are not subjected to the second level of decomposition.

- Five narrowband components are decomposed for a second time according to the BER criterion, and the residual components are decomposed according to $K = 2$.

- Five narrowband components are decomposed for a second time according to the BER criterion, and the residual components are decomposed according to $K = 3$.

D. Third decomposition according to the BER criterion

- Five narrowband components are decomposed three times according to the BER criterion, and the residual components are not subjected to the second level of decomposition.

- Five narrowband components are decomposed three times according to the BER criterion, and the residual components are decomposed according to $K = 2$.

- Five narrowband components are decomposed three times according to the BER criterion, and the residual components are decomposed according to $K = 3$.

- Decomposing all components three times according to the BER criterion.

E. Third decomposition and removal of secondary residual components

- Based on the optimal decomposition, the secondary residual components are removed.

**Experimental results and analysis.** 1. Comparison Experiment I

Group A and Group B are measured by three error indicators and compared with the first level of decomposition. The forecasting results of each model are shown in Table 7.

The results in Table 7 show the following:

- The experimental errors of direct secondary decomposition and secondary decomposition according to the central frequency are better than those of first-level decomposition in most instances, but there are also cases where the accuracy is extremely poor. These results indicate that deeper levels of decomposition do not equate to more accurate results.

- In the experiments conducted on both the 2020 and 2021 datasets, the mode number of Group B is more than that in Group A. In the experiments based on LSTM, the error of

**Table 7. Forecasting results of Group A and Group B.**

| Indicator | Model | 2020 | | | 2021 | | |
|---|---|---|---|---|---|---|---|
| | | First level | Group A | Group B | First level | Group A | Group B |
| MAE | LSSVM | 573.1176 | 197.3233 | 190.7865 | 489.8285 | 320.3984 | 320.3929 |
| | ELM | 1958.7979 | 32688.73 | 4056138 | 1300.4613 | 587.2526 | 1465.8817 |
| | DBN | 499.3744 | 244.252 | 202.6702 | 431.1148 | 225.0884 | 213.6698 |
| | LSTM | 3916.722 | 3912.2559 | 1023.958 | 1774.359 | 992.1692 | 625.4075 |
| RMSE | LSSVM | 706.9316 | 255.4071 | 247.16437 | 631.1439 | 403.0863 | 404.7089 |
| | ELM | 2576.0908 | 41155.958 | 5326.8536 | 1561.6901 | 881.4671 | 1882.439 |
| | DBN | 623.8845 | 314.3993 | 279.9758 | 548.7777 | 300.3513 | 293.8501 |
| | LSTM | 4243.249 | 3998.325 | 1304.2325 | 2190.5804 | 1168.814 | 768.2987 |
| MAPE(%) | LSSVM | 0.7596 | 0.2613 | 0.2523 | 0.6549 | 0.4267 | 0.4276 |
| | ELM | 2.5915 | 43.4915 | 1465.8817 | 1.7603 | 0.7773 | 1.9743 |
| | DBN | 0.619 | 0.3591 | 0.2651 | 0.5756 | 0.2997 | 0.2839 |
| | LSTM | 5.2461 | 5.2043 | 1.3706 | 2.344 | 1.3429 | 0.8397 |

Group B has markedly decreased compared with that of Group A. However, in the experiments based on other models, the advantage of Group B is not large, and there is even one large error. Thus, more modes do not equate to less error with the same decomposition levels. In addition, both decomposition methods have certain drawbacks and great limitations in reducing the forecast accuracy.

2. Comparison experiment II

In the experiments based on the BER criterion, only some subsignals satisfy the condition of the third decomposition. Therefore, in addition to the experimental results of Group C and Group D, another comparison experiment is set up for all the subsignals obtained by the second decomposition to be decomposed for the third time to verify the accuracy and superior performance of the recursive decomposition method based on the BER criterion. Experimental results are shown in Table 8.

The errors of Group D and the comparison data in Table 8 show that a deeper decomposition without satisfying the BER criterion is likely to cause an error explosion. The error

**Table 8. Experimental error results for the third and fourth groups.**

| Indicator | Model | 2020 | | | 2021 | | |
|---|---|---|---|---|---|---|---|
| | | Group C | Group D | Comparative data | Group C | Group D | Comparative Data |
| MAE | LSSVM | 204.9825 | 159.527 | 161.651 | 315.0771 | 251.2095 | 253.226 |
| | ELM | 1709.5477 | 1595.1924 | 1951437 | 644.626 | 558.1538 | 1235.4259 |
| | DBN | 286.868 | 229.1196 | 278.1702 | 290.0355 | 225.9414 | 259.521 |
| | LSTM | 3627.924 | 3592.7705 | 75903 | 978.1682 | 882.723 | 1697.3942 |
| RMSE | LSSVM | 263.7048 | 193.667 | 197.4948 | 396.1654 | 321.8135 | 323.3518 |
| | ELM | 2245.8487 | 1969.1107 | 2200894 | 789.1289 | 686.8216 | 1491.9478 |
| | DBN | 332.7216 | 267.1993 | 356.7122 | 359.461 | 288.6117 | 326.7534 |
| | LSTM | 3705.5479 | 3676.57 | 75908 | 1142.9699 | 1100.5448 | 1974.204 |
| MAPE(%) | LSSVM | 0.2718 | 0.2122 | 0.2153 | 0.4197 | 0.3352 | 0.338 |
| | ELM | 2.2741 | 2.0964 | 2584.696 | 0.87 | 0.7536 | 1.6566 |
| | DBN | 0.3796 | 0.3037 | 0.3504 | 0.3853 | 0.2996 | 0.3444 |
| | LSTM | 4.8301 | 4.7808 | 100.702 | 1.328 | 1.1768 | 2.3134 |

**Table 9. Percentage reduction for secondary and tertiary decomposition.**

| Year | Decomposition Level | LSSVM | ELM | DBN | LSTM |
|---|---|---|---|---|---|
| 2020 | 2 | -64.23% | -12.72% | -42.55% | -7.37% |
| | 3 | -72.17% | -18.56% | -54.12% | -8.27% |
| 2021 | 2 | -35.68% | -50.43% | -32.72% | -44.87% |
| | 3 | -48.71% | -57.08% | -47.59% | -50.25% |

comparing Group C and Group D following the BER criterion with the first-level decomposition is shown in Table 9 and Fig 8. Experimental error decreases to some extent for each level of decomposition for all forecasting models. Combined with the previous conclusion that the number of decomposition levels and modes is not proportional to the reduction in forecasting error, the superiority of the BER criterion in decomposing time series is further shown.

3. Comparison experiment III

To describe the influence of the residual components on the forecast accuracy, the secondary residual component with the largest PE is chosen to be discarded as a comparison experiment. The relative percentage decrease in the error for Group E compared with Group D is shown in Table 10.

As shown in Table 10, the errors of the four groups of experiments decreased to a certain extent after the removal of the secondary residual components. Fig 9 shows the average percentage of MAE deletion. The experiments with LSSVM as the forecasting model show larger decreases after removing the residual components, indicating that the decomposition of wind power generation series is more accurate in this experiment, and the separation of noise in the series is more successful.

**Summary.**   These comparative experiments and data results show the following:

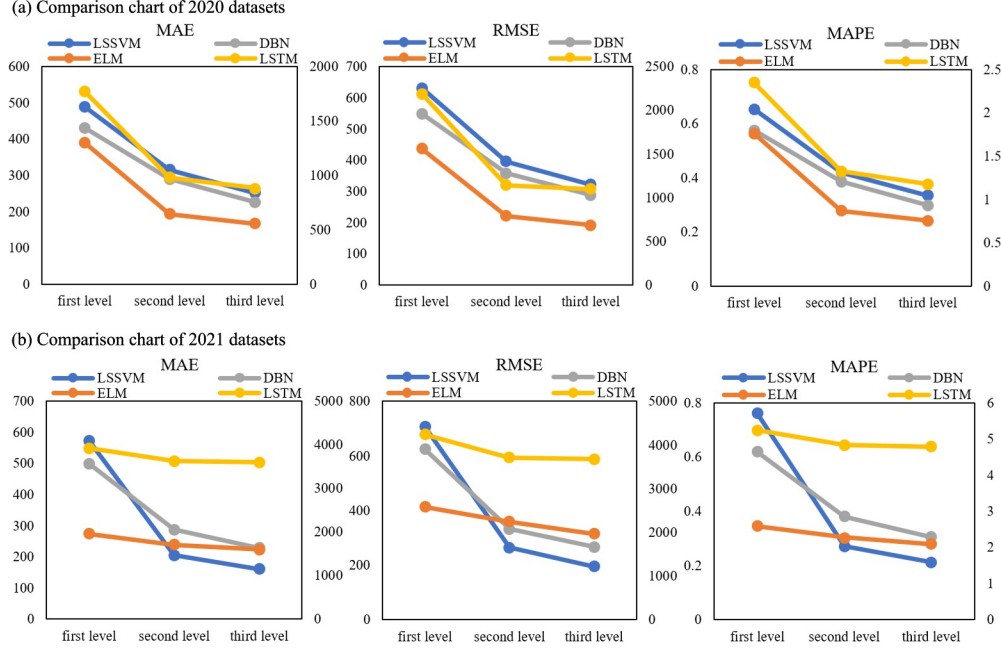

(a) Comparison chart of 2020 datasets

(b) Comparison chart of 2021 datasets

**Fig 8. Comparison of experimental errors in three levels.**

**Table 10. Percentage decrease in the error of discarding secondary residual components.**

| Models | 2021 | | | 2021 | | |
|---|---|---|---|---|---|---|
| | MAE | RMSE | MAPE | MAE | RMSE | MAPE |
| *LSSVM* | -52.19% | -48.64% | -52.32% | -75.06% | -76.11% | -74.98% |
| *ELM* | -6.10% | -1.05% | -6.08% | -18.66% | -20.72% | -18.60% |
| *DBN* | -44.11% | -42.68% | -43.91% | 1.92% | -12.37% | 3.65% |
| *LSTM* | -0.29% | -0.86% | -0.51% | -10.08% | -11.97% | -9.95% |

- To achieve better separation for higher forecasting accuracy, blind decomposition is undesirable. Neither deep decomposition levels nor a large number of modes is equivalent to a small error.

- Traditional decomposition methods are more random in terms of validity and effectiveness, which is inappropriate as a criterion for judging the mode number.

- Recursive decomposition based on BER has a complete mathematical derivation process and shows stability in the real forecasting process, which is more objective than other decomposition methods.

## Conclusions

In this paper, we propose a recursive decomposition method based on the branch error reduction criterion to decompose wind power generation into more regular and easily trained

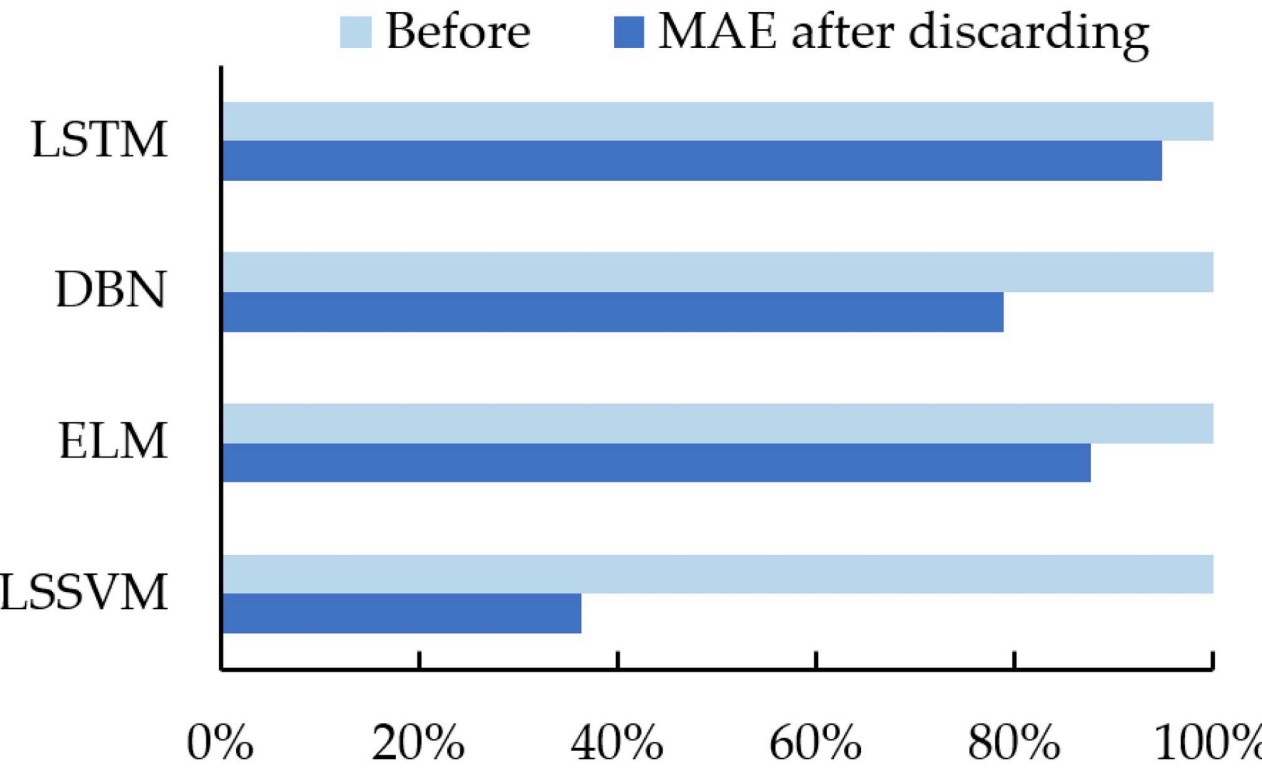

**Fig 9. Average percentage of MAE after removing the secondary residual components.**

multiple modes. Four forecasting models, LSSVM, ELM, DBN, and LSTM, are used for forecasting, and the superior performance of the proposed decomposition method is primarily shown in the following results:

- Taking full advantage of VMD, the proposed method can decompose the original time series into subcomponents with more easily captured features.

- The branch error reduction criterion is supported by mathematical theory, which improves the reliability and robustness of the overall model.

- The ambiguous judgment method is abandoned, and a mathematical guideline is used to facilitate program integration and modular design of signal decomposition.

Because the trend of power generation and the influencing factors change over time, the distribution of the dataset used for training is not consistent with the new data, resulting in the previous model not being able to forecast the present data at a high level of accuracy, which means there is a distribution drift phenomenon. To improve model generalizability and make the distribution of training and testing data as consistent as possible, the distribution drift will be improved in the future based on the existing research using the sample weighting strategy to improve prediction accuracy.

## Supporting information

**S1 Data.**
(CSV)

## Author Contributions

**Conceptualization:** Xiao Li, Junhong Ni.

**Data curation:** Fen Xiao, Siyu Yang.

**Formal analysis:** Xiao Li.

**Funding acquisition:** Fen Xiao, Siyu Yang.

**Methodology:** Junhong Ni.

**Project administration:** Fen Xiao, Siyu Yang.

**Resources:** Fen Xiao, Siyu Yang.

**Supervision:** Fen Xiao, Siyu Yang.

**Validation:** Xiao Li.

**Writing – original draft:** Xiao Li.

**Writing – review & editing:** Xiao Li, Junhong Ni.

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
