## [Decision Letter · Decision Letter 0]

9 Jan 2024

PONE-D-23-26209Branch Error Reduction Criterion-Based Signal Recursive Decomposition and Its Application to Wind Power Generation ForecastingPLOS ONE

Dear Dr. li,

Thank you for submitting your manuscript to PLOS ONE. After careful consideration, we feel that it has merit but does not fully meet PLOS ONE’s publication criteria as it currently stands. Therefore, we invite you to submit a revised version of the manuscript that addresses the points raised during the review process.

Please submit your revised manuscript by Jan 14 2024 11:59PM. If you will need more time than this to complete your revisions, please reply to this message or contact the journal office at plosone@plos.org. Please include the following items when submitting your revised manuscript:A rebuttal letter that responds to each point raised by the academic editor and reviewer(s). You should upload this letter as a separate file labeled 'Response to Reviewers'.A marked-up copy of your manuscript that highlights changes made to the original version. You should upload this as a separate file labeled 'Revised Manuscript with Track Changes'.An unmarked version of your revised paper without tracked changes. You should upload this as a separate file labeled 'Manuscript'.If applicable, we recommend that you deposit your laboratory protocols in protocols.io to enhance the reproducibility of your results. Protocols.io assigns your protocol its own identifier (DOI) so that it can be cited independently in the future. For instructions see: https://journals.plos.org/plosone/s/submission-guidelines#loc-laboratory-protocols. Additionally, PLOS ONE offers an option for publishing peer-reviewed Lab Protocol articles, which describe protocols hosted on protocols.io. Read more information on sharing protocols at https://plos.org/protocols?utm_medium=editorial-email&utm_source=authorletters&utm_campaign=protocols.

We look forward to receiving your revised manuscript.

Kind regards,

Samuel Asante Gyamerah, Ph.D

Academic Editor

PLOS ONE

Journal Requirements:

"Funding information:

State Grid Fujian Electric Power Co. Ltd.: SGTYHT/20-JS-223(SGFJJY00GHJS2200054)

We note that one or more of the authors have an affiliation to the commercial funders of this research study : State Grid Fujian Electric Power Co. Ltd

(2) Please also provide an updated Competing Interests Statement declaring this commercial affiliation along with any other relevant declarations relating to employment, consultancy, patents, products in development, or marketed products, etc.  

Within your Competing Interests Statement, please confirm that this commercial affiliation does not alter your adherence to all PLOS ONE policies on sharing data and materials by including the following statement: ""This does not alter our adherence to  PLOS ONE policies on sharing data and materials.” (as detailed online in our guide for authors http://journals.plos.org/plosone/s/competing-interests). If this adherence statement is not accurate and  there are restrictions on sharing of data and/or materials, please state these. 

Please note that we cannot proceed with consideration of your article until this information has been declared.

"This research was fully funded by the scientific and technological project of State Grid 

Fujian Electric Power Co. Ltd. (SGTYHT/20-JS-223(SGFJJY00GHJS2200054)).The 

authors would like to express their gratitude to AJE for the expert linguistic services 

provided."

Funding information should not appear in the Acknowledgments section or other areas of your manuscript. We will only publish funding information present in the Funding Statement section of the online submission form. 

"Funding information:

State Grid Fujian Electric Power Co. Ltd.: SGTYHT/20-JS-223(SGFJJY00GHJS2200054)

Reviewers' comments:

Reviewer's Responses to Questions

**Comments to the Author**

1. Is the manuscript technically sound, and do the data support the conclusions?

Reviewer #1: Yes

Reviewer #2: Yes

2. Has the statistical analysis been performed appropriately and rigorously? 

Reviewer #1: Yes

Reviewer #2: Yes

3. Have the authors made all data underlying the findings in their manuscript fully available?

Reviewer #1: No

Reviewer #2: Yes

4. Is the manuscript presented in an intelligible fashion and written in standard English?

Reviewer #1: Yes

Reviewer #2: Yes

5. Review Comments to the Author

Reviewer #1: This is a very good study which is novel. Hybrid methods are known to perform better than the underlying or single methods to analyse data. The authors also presented the same work with some level of great detail into the suggested method. However, there are few corrections that may need to be done. In line 18 its written In Ref. [6] which i presume was supposed to write the author's name like what the authors did in line 19. This needs correction. Similar mistake is made in line 24 and 43.

Decomposition has been used as a preprocessing method in many studies as also cited in this paper. I suggest adding one or 2 sentences after paragraph 1 in Introduction justifying why the wind generation data needs to be decomposed. In Line 192 I suggest carrying out stationarity Tests such as the Augmented Dickey-Fuller (ADF) test and the Kwiatkowski-Phillips-Schmidt-Shin (KPSS) test and then make a conclusion that the data is non-stationary.

It would be interesting to add few more years instead of using only 2 years to carry out a study.

Reviewer #2: The motivation for the study has been well explained. Choice of Branch Error reduction over VMD has been duly justified. Recommendation for practice and further studies have been given. Some statements in the introductory section need to be cited

6. PLOS authors have the option to publish the peer review history of their article (what does this mean?). If published, this will include your full peer review and any attached files.

Reviewer #1: No

Reviewer #2: **Yes: **EMMANUEL NUMAPAU GYAMFI

---

## [Author Response · Author response to Decision Letter 0]

25 Jan 2024

To Academic Editor

Thank you so much for your professional and valuable comments on our manuscripts, we have made changes and improvements accordingly.

Comments: 

Reply: Thank you so much for the kind suggestion. We have carefully checked our manuscript to ensure that this manuscript meets PLOS ONE's style requirements.

Reply: Thank you very much for your reminder. The code has been packaged for easy running.

3. The Funding Statement and Competing Interests Statement need to be updated.

Reply: Thank you for your kind suggestion. The Funding Statement and Competing Interests Statement are updated.

4. Funding information should not appear in the Acknowledgments section or other areas of your manuscript. We will only publish funding information present in the Funding Statement section of the online submission form. Please include your amended statements within your cover letter; we will change the online submission form on your behalf.

Reply: Sorry for the oversight, the funding information has been placed in Funding Statement section of the online submission form.

The updated Funding Statement is as the following:

Funding information: State Grid Fujian Electric Power Co. Ltd.: SGTYHT/20-JS-223(SGFJJY00GHJS2200054). The funders had no role in study design, data collection and analysis, decision to publish, or preparation of the manuscript. The funder provided support in the form of salaries for authors Fen Xiao and Siyu Yang, but did not have any additional role in the study design, data collection and analysis, decision to publish, or preparation of the manuscript. The specific roles of these authors are articulated in the ‘author contributions’ section.

5. In your Data Availability statement, you have not specified where the minimal data set underlying the results described in your manuscript can be found.

Reply: Thank you for your suggestion. The minimal data set used in the manuscript has been uploaded as Supporting Information.

6. Please review your reference list to ensure that it is complete and correct.

Reply: Thank you for the kind suggestion. All references have been checked carefully. Four new references have been supplemented.

To Reviewers #1

We really appreciate the professional and valuable comments on our manuscript. We have made the revisions and improvements accordingly.

1. In line 18 its written In Ref. [6] which i presume was supposed to write the author's name like what the authors did in line 19. This needs correction. Similar mistake is made in line 24 and 43.

Reply: Thank you for the kind suggestion. The citations formats have been corrected accordingly.

2. Decomposition has been used as a preprocessing method in many studies as also cited in this paper. I suggest adding one or 2 sentences after paragraph 1 in Introduction justifying why the wind generation data needs to be decomposed. 

Reply: Thank you for the professional comment. The reason for the decomposition of the generation time series is supplemented at the beginning of the 4th paragraph in the introduction section: “The time series of electricity generation is generally a broadband signal, and its future trend is not stable. Therefore, it is difficult to approximate the relationship between historical measurements and its future changes. The future trend of a narrowband signal is normally considered to be more stable. Therefore, the second type of hybrid model is used to decompose the time series of power generation into narrowband modes, and the final forecasted results are obtained by summarizing the forecasted results of each mode” (Line 37-43).

3. In Line 192 I suggest carrying out stationarity Tests such as the Augmented Dickey-Fuller (ADF) test and the Kwiatkowski-Phillips-Schmidt-Shin (KPSS) test and then make a conclusion that the data is non-stationary.

Reply: Thank you for the enlightening comment.

The ADF test results of partial and total generation time series of 2020 and 2021 are supplemented to Table 2 in the revised manuscript. Detailed statistics of the ADF test are also presented in Table R.1, which indicates the non-stationarity of the time series data. Due to limited space, can we just not show Table R.1 in the revised manuscript, thank you.

Table R.1. ADF test results and critical values (in 'Response to Reviewers.docx')

4. It would be interesting to add few more years instead of using only 2 years to carry out a study.

Reply: Thank you for the practically significant suggestion.

The historical data of power generation in Fujian Province from Jan 1 2012 to Jun 30 2022 is shown in Fig. R.1, with a total of 3834 daily power generation data. A total of 3653 power generation data in the decade 2012- 2021 is selected as the training set, and a total of 181 power generation data in the first six months of 2022 is selected as the testing set. The same comparison experiments are conducted against the manuscript and the results are shown in Table R.2. 

Fig. R.1 Power generation trend of Fujian Province from Jan o1 2012 to Jun 30 2019 (in 'Response to Reviewers.docx')

Table R.2. Comparative experiments (in 'Response to Reviewers.docx')

According to Table R. 2, the same conclusions can be drawn: 

 a. More decomposition layers or more decomposition numbers cannot be equated with better decomposition effect; 

 b. There are randomness and limitations in the final effect of direct decomposition and decomposition according to whether the center 

 frequency is aliased; 

 c. Recursive decomposition based on BER performs more consistently and efficiently in the experiment.

Because the trend of power generation and the influencing factors change over time, the distribution of the dataset used for training is not consistent with the new data, resulting in the previous model not being able to forecast the present data at a high level of accuracy, which indicates that the distribution drift occurs.

Although the same conclusions can be drawn, the electricity generation data have several characteristics: they are highly influenced by policy, data distribution evolves over time. Compared with one year's data, too much experimental data seems to be less suitable for comparison and presentation to some extent. Therefore, would it be more succinct to use only single year data in the experiment? We look forward to your valuable suggestions.

To Reviewers #2

The authors would like to thank you for taking the time to read our manuscript. We have modified our manuscript according to the suggestions and comments. 

1. Some statements in the introductory section need to be cited.

Reply: Thank you for the kind suggestion. In the introduction section, three references are supplemented, i.e., Ref [1], Ref [2] and Ref [4]. In the comparative experiments section, Ref [25] is supplemented.

---

## [Decision Letter · Decision Letter 1]

20 Feb 2024

Branch Error Reduction Criterion-Based Signal Recursive Decomposition and Its Application to Wind Power Generation Forecasting

PONE-D-23-26209R1

Dear Dr. li,

We’re pleased to inform you that your manuscript has been judged scientifically suitable for publication and will be formally accepted for publication once it meets all outstanding technical requirements.

Kind regards,

Samuel Asante Gyamerah, Ph.D

Academic Editor

PLOS ONE

Additional Editor Comments (optional):

Reviewers' comments:

Reviewer's Responses to Questions

**Comments to the Author**

1. If the authors have adequately addressed your comments raised in a previous round of review and you feel that this manuscript is now acceptable for publication, you may indicate that here to bypass the “Comments to the Author” section, enter your conflict of interest statement in the “Confidential to Editor” section, and submit your "Accept" recommendation.

Reviewer #1: All comments have been addressed

2. Is the manuscript technically sound, and do the data support the conclusions?

Reviewer #1: Yes

3. Has the statistical analysis been performed appropriately and rigorously? 

Reviewer #1: Yes

4. Have the authors made all data underlying the findings in their manuscript fully available?

Reviewer #1: Yes

5. Is the manuscript presented in an intelligible fashion and written in standard English?

Reviewer #1: Yes

6. Review Comments to the Author

Reviewer #1: (No Response)

7. PLOS authors have the option to publish the peer review history of their article (what does this mean?). If published, this will include your full peer review and any attached files.

Reviewer #1: **Yes: **Willard Zvarevashe

---

## [Editor Report · Acceptance letter]

12 Mar 2024

PONE-D-23-26209R1 

PLOS ONE

Dear Dr. Li, 

I'm pleased to inform you that your manuscript has been deemed suitable for publication in PLOS ONE. Congratulations! Your manuscript is now being handed over to our production team.

Kind regards, 

on behalf of

Dr. Samuel Asante Gyamerah 

Academic Editor

PLOS ONE